# Measurement and Modeling of the Optical Properties of Adipose Tissue in the Terahertz Range: Aspects of Disease Diagnosis

**DOI:** 10.3390/diagnostics12102395

**Published:** 2022-10-01

**Authors:** Irina Y. Yanina, Viktor V. Nikolaev, Olga A. Zakharova, Alexei V. Borisov, Konstantin N. Dvoretskiy, Kirill V. Berezin, Vyacheslav I. Kochubey, Yuri V. Kistenev, Valery V. Tuchin

**Affiliations:** 1Institute of Physics, Saratov State University, 410012 Saratov, Russia; 2Laboratory of Laser Molecular Imaging and Machine Learning, Tomsk State University, 634050 Tomsk, Russia; 3Department of Medbiophysics, Saratov State Medical University, 410012 Saratov, Russia; 4Institute of Precision Mechanics and Control, FRC “Saratov Scientific Centre of the Russian Academy of Sciences”, 410028 Saratov, Russia

**Keywords:** terahertz spectroscopy, molecular modeling, absorption spectra, adipose tissue, oleic acid, water, dehydration

## Abstract

In this paper, the measurement and modeling of optical properties in the terahertz (THz) range of adipose tissue and its components with temperature changes were performed. Spectral measurements were made in the frequency range 0.25–1 THz. The structural models of main triglycerides of fatty acids are constructed using the B3LYP/6-31G(d) method and the Gaussian03, Revision B.03 program. The optical density (OD) of adipose tissue samples decreases as temperature increases, which can be associated mostly with the dehydration of the sample. Some inclusion of THz wave scattering suppression into the OD decrease can also be expected due to refractive index matching provided by free fatty acids released from adipocytes at thermally induced cell lipolysis. It was shown that the difference between the THz absorption spectra of water and fat makes it possible to estimate the water content in adipose tissue. The proposed model was verified on the basis of molecular modeling and a comparison with experimental data for terahertz spectra of adipose tissue during its heating. Knowing the exact percentage of free and bound water in adipose tissue can help diagnose and monitor diseases, such as diabetes, obesity, and cancer.

## 1. Introduction

Terahertz (THz) spectroscopy allows one to determine the complex refractive index of the medium under study, which is important for creating a functional THz tomography with high sensitivity to changes in the concentration of metabolites and accurate marking of the boundaries of the pathological lesions. Therefore, the development of spectroscopic methods for studying biological tissues in the THz frequency range, providing detection and visualization of metabolic and pathological processes, has caused great interest in recent years, especially as an additional channel for obtaining information in multimodal systems in combination with the other approaches, such as using biosensors [1] and optical coherence tomography, or polarized light imaging [2]. The contrast between healthy and diseased tissues for THz wave probing is due to differences in their water content and degree of tissue dehydration, for wax embedded tissue samples [3], as well as in optical properties of tissues, such as muscle [4], liver [5], colon [6], and skin [7], and their structures.

THz waves can be utilized in spectroscopy and imaging in both transmission and reflection modes [8]. In THz reflectance imaging, depth information is obtained using the time delay and amplitude of the registered reflected wave. This method is used to determine the absorbance and phase shift by different types of tissues (normal tissues and tumors) [9,10]. The penetration of THz waves into issues, depending on the amount of fat and water content, can vary in the range of a few microns to a few millimeters due to the absorption by polar molecules [11]. Although computed tomography (CT), magnetic resonance imaging (MRI), and fluorescence imaging are widely used to identify the boundary between healthy and malignant tissues, for some brain cancers, such as glioma, tumor margins are not well detectable. The brain is a very high-lipid organ, and the amount of lipid causes high contrast in THz imaging (THI). Cancerous tumors have higher protein and lower lipid levels than healthy tissue, and the proteins are highly absorbing in the THz spectral region. By performing THI in vivo, it was observed that the tumor border was clearly determined [12]. Therefore, neurosurgeons can use THI in surgery to obtain a high contrast image.

THz signals with a wavelength of approximately 300 µm provide a good balance between penetration, deepness, and spatial resolution to identify a molecular biomarker spatial distribution in tissue [13]. Biomarker detection can be helpful in the diagnosis of cancer. As different substances have different THz spectra (i.e., spectral “fingerprints”), THz spectroscopy (THS) can be applied to realize fast and accurate identification of biomarkers in cancer tissue. Furthermore, THS can be combined with various algorithms to realize the quantitative analysis of cancer biomarkers, which may be a potential tool for rapid cancer staging [14]. However, in most cases, various substances, such as water, proteins, lipids, and other organic components, are also present in the tissue, and the cancer biomarker concentration is usually very low. These conditions result in a low spectral signal-to-noise ratio (SNR), which gives rise to absorption peaks that cannot be readily identified [15]. Therefore, current THz-based studies of biomarkers are mostly performed on pure biomarkers. If we can construct THz images based on the resonance peaks of biomarkers, cancerous areas can be identified more accurately. However, further development is required to realize this objective as current imaging systems with high-power continuous-wave THz sources can only generate THz waves at a fixed frequency, and pulsed THz sources cannot provide enough spectral power. Thus, both THI and THS are limited by source energy, leading to poor SNR in the spectral results. To address this issue, some researchers are working to enhance the spectral SNR, which is expected to help detection of biomarkers in mixed samples. THS can also accurately diagnose brain tissue lipid deficiency [16,17]. It should be noted that the white matter has lots of myelin, whose sheath is a multi-layered membrane, mainly consisting of lipids, whereas the gray matter of the brain has a higher content of water and proteins, which provides higher signal intensities in THz images than the tissues with high lipid content.

The polarization sensitive optical imaging and THI may be combined to provide useful information for the differentiation of healthy and cancerous tissues (nonmelanoma skin cancer [18,19], colon cancer [20]). There is a difference in bound and free water contents between normal and cancerous tissues, such as benign: seborrheic keratosis, pigmented nevi; malignant: malignant melanoma, basal cell carcinoma [21], tumor bearing tissue from rat livers [22]. Due to higher water content, cancer exhibits higher absorption relative to normal skin and therefore leads to a lower remitted signal and consequently lower reflectivity of cancerous areas [23].

The presence of water in adipose tissue is a sign of diabetes [24], obesity [25], and cancer [26]. The water content in adipose normal and pathological specimens was seven to ten times as great as the protein content [27]. Almost all the remainder of the fat pads consists of lipids. Normal abdominal adipose tissue of mice contains 84–91% fat and 8–14% water. In diseased tissue, the water content may increase to more than 30% with simultaneous reduction of the fat content. Since cancer cachexia promotes an increase in body water content, researchers speculated that the enlargement mesenteric and retroperitoneal adipocytes was caused at least in part by water retention, as opposed to an increase in lipid [28,29]. The complex anatomic structure of the breast, represented by an association of closely juxtaposed fat and glandular and fibrous connective tissues, breast cancers are always surrounded by tissues with heterogeneous conductivity [30]. For example, the main component of the breast in young patients is breast gland, with little adipose tissue. However, the main component is adipose tissue in old patients. Furthermore, the percentage of water in breast carcinoma is more than that in breast tissue and adipose tissue [31]. 

Lipids attenuate THz radiation less strongly than polar molecules. The absorption rate for all lipids increases with frequency and reaches a maximum for about 2 THz [32,33,34,35]. The difficulty of interpreting results of measurements and the transition from these measurements to in vivo diagnostics is caused by an uncontrolled environment, e.g., diffusion into a sample of saline during tissue storage, changes in the level of hydration during the measurement, effects of scattering, etc. [36]. 

Choe et al. [37] demonstrated in vivo the distribution of different water types (i.e., tightly hydrogen bound, strongly hydrogen bound, weakly hydrogen bound and unbound) in the human stratum corneum (SC) which is rich in lipids. Strongly bound water (double donor–double acceptor, DDAA–OH) and weakly bound water (single donor–single acceptor, DA–OH) were shown to represent more than 90% of the entire water content of SC, while tightly bound water (single donor–double acceptor, DAA) and free water molecule types represent the remaining < 10%.

The absorption coefficients of skin dermis and epidermis are given as 70% and 20% of the absorption coefficient of water, respectively [38]. For subcutaneous tissues, approximately 40% and 60% of absorption coefficient of lipids and water are characteristic, respectively [39,40]. It is important to notice that at ex vivo skin topical application of the hyperosmotic optical clearing agent (OCA), free water, and weakly bound water are displaced, causing tissue dehydration [41]. Enhanced free water content in SC can be provided at increased humidity of the environment [42].

Approximately 60–85% of the weight of white adipose tissue is lipid, with 90–99% being triglyceride. Small amounts of free fatty acids, diglyceride, cholesterol, and phospholipid and minute quantities of cholesterol ester and monoglyceride are also present. In this lipid mixture, six fatty acids make up approximately 90% of the total, and these are myristic, palmitic, palmitoleic, stearic, oleic, and linoleic acids. The remaining weight of white adipose tissue is composed of water (5–30%) and proteins (2–3%) [43]. For example, for adipose tissue in mesenteric and subcutaneous depots, total water volume was 14 ± 1.4% with extracellular component of 11 ± 1.1% [44].

The SC can be used as a model for adipose tissue water balance prediction as it contains a large amount of lipids [37], normally of about 30% in the upper layers with rest of proteins and water [41]. The protein-to-lipid ratio in the adipose tissue is one of the important parameters [45].

Guo et al. [46] demonstrated that the THz digital holographic imaging system can be utilized to investigate natural dehydration processes in adipose tissue. The authors showed that from THz images of biological specimens, distinctive water content as well as dehydration features of adipose tissues can be obtained. As shown in the paper, the degree of dehydration of porcine samples was about 70–80%. The experimental results imply that dehydration features of adipose tissues in different animal bodies have some discrepancies, including the decay time constant and variation extent of THz absorption.

The external mechanical pressure on the biological tissue can cause free water to come out of the tissue first and then bound water [38]. Osmotic pressure acts in a similar way, and it leads to the loosening of weakly bound water. Presumably, the picture in adipose tissue should be qualitatively similar, the differences can be at the percentage level. Our hypothesis is that adipose tissue heating can lead to similar processes with free and bound water. Namely, to cause tissue dehydration.

The main goal of this study is to create a model of absorption properties of adipose tissue in the THz range, allowing for analysis of the role of free and bound tissue water and its comparison with experimental data received for different tissue temperatures. The hydration model of adipose tissue is based on a quantum-mechanical atomistic simulation method in the framework of the density functional theory (DFT), which allows one to compute a wide variety of properties of almost any kind of atomic system including tissue molecular structures [47]. Moreover, we aimed to show that the proposed model of fat with different ratios of free and bound water can be considered as a model for the various pathological conditions of adipose tissue.

An addition, we present a brief review of experimental data for the absorption and dispersion of adipose tissues and their components in the THz range.

## 2. Materials and Methods

### 2.1. Materials

Abdominal porcine adipose tissue samples were used in this study. A total of eight samples were investigated. The thickness of the samples was approximately 1.5 mm, and their area amounted to 1 cm^2^. In advance, samples of adipose tissue frozen at a temperature of −20 °C were cut into pieces with a thickness slightly above 2 mm. Then, using a cylindrical punch with an inner diameter of 9 mm, a cylindrically shaped sample was cut. This sample was thawed, placed inside a metallic holder with a fixed height of 1.25 mm, and excess adipose tissue was removed by a scalpel. To evaluate THz wave attenuation (absorption) coefficient, rather precise knowledge of sample thickness is needed. Therefore, thickness measurements were provided for each sample placed between two glass slides, and measurements were performed at several points of the sample. Metallic holder height measurements were measured using the micrometer “MK 0–25 mm”, model 102 (Plant “Caliber”, Russia). The error of each measurement was approximately 10 μm. The obtained thicknesses were averaged.

OD of H_2_O and oleic acid at different thicknesses was measured in THz range at room temperature. Oleic acid was chosen because its percentage in adipose tissue is the highest (45%) [48].

### 2.2. Methods of Measurement

To monitor changes in adipose tissue, the temperature was varied from 25 °C to 70 °C in increments of 1 °C. For the heating of samples, a laboratory DC power supplier (YIHUA-305D) was used, the heating ability of which was controlled by changing the applied voltage (Figure 1a). The dependence of the temperature inside the sample holder on the current passed through the heater (nichrome resistor wire gauge) was obtained for reliable measurements (see Figure 1c). Under normal conditions, a thermocouple was placed inside the sample holder with tissue, and the electric current was slowly raised with a step of 0.2 A. At each step, a time interval of 2 min was maintained, sufficient for the temperature inside the sample holder to stop changing. This approach made it possible to minimize the effects of a temperature gradient.

A prototype of a heating element was developed (Figure 1b) consisting of a metal base (1) with a thickness of 2 mm, in which a through round hole 1 cm in diameter was made. Taking into account the diameter of the laser beam (3.5 mm), the hole diameter is sufficient for spatial scanning in several points. The base included two parts. The first plate was square, with a hole for the laser beam to pass through. The second plate was similar to the first, but with the presence of a protrusion, namely a leg (2) for fixing on the table of the THz spectrometer. Fluoroplastic (3) 2 × 2 cm in size and 0.5 mm thick was glued to the bases for tightness. A washer (4) with an inner diameter of 1.5 cm and a thickness of 1.25 mm was placed in the center of the fluoroplast, which serves as a sample holder. The measurement accuracy of the washer thickness was 10 µm. The inner hole of the washer was 9 mm. It also had a 0.5 mm slot for wires. A wire (the number of turns was 2) (5) approximately 9 cm long was placed along the inner edge of the hole in the washer in a spiral shape. Both ends were connected to wires through terminals connected to a laboratory DC power supply. The tungsten wire was coated with a special thermally conductive electrical insulating varnish. The wire was insulated from the sample. At the edges, the metal bases were pulled together with four screws (6) and tightened with nuts.

THz spectral measurements were made using a real-time T-SPEC terahertz spectrometer (EKSPLA, Vilnius, Lithuania) working in the frequency range 0.25–1 THz with a software THz Spectrometer 2D. A photoconductor antenna illuminated by ultrashort laser pulses was used for the generation of THz radiation and its detection. The pumping laser provided pulses of 10–150 fs at 1050 ± 40 nm wavelength with power of about 100 mW and 30–100 MHz pulse repetition rate. For more efficient collimation and focusing of THz radiation, a substrate lens fabricated from high resistance silicon was attached to the backside of each antenna. The sample holder was placed in the optical path. Atmospheric air was in the optical path, and the cover of the device was open to ensure the outflow of excess heat from the outer part of the sample holder. There were no advantages in using the nitrogen in the range from 0.2 to 1 THz compared to atmospheric air. The reference THz signal was the signal of THz wave passed through the cuvette without a sample, i.e., through two fluoroplastic plates, each 0.5 mm thick.

The cell with the sample without a thermocouple was placed in the optical path of the T-SPEC spectrometer and measurements were carried out as follows:The THz spectrum was measured at room temperature at 4 points with a vertical and horizontal step of 0.4 mm. This was performed by moving the cuvette by means of a stepper motor.Voltage was applied to raise the temperature by 5 °C according to the calibration curve (Figure 1c). Time was kept for more than 2 min. The THz spectrum was recorded at 4 points with a vertical and horizontal step of 0.4 mm.The voltage was raised, and the next temperature point was taken.

### 2.3. Methods of Modelling

The structural models of five triglycerides of fatty acids (oleic, linoleic, palmitic, stearic, α-linolenic) are constructed using B3LYP/6-31G(d) method and the Gaussian03, Revision B.03 program from [49]. The vibrational wavenumbers and intensities in the IR spectra were calculated. The molecular model of porcine fat was constructed basing on five models of triglycerides of fatty acids. The IR spectra of porcine fat are simulated using the supermolecular approach. The content of these fatty acid triglycerides in the models is shown in Table 1. The halfwidth of all Lorentzian profiles was taken to be 10 cm^−1^. For better agreement with the experiment, the calculated vibrational wavenumbers were corrected using linear frequency scaling [50].

## 3. Results

### 3.1. Review of the Optical Properties of Water, Lipids and Fatty Acids in the THz Range

Refractive indices and absorption coefficients of water [2,51,52,53,54,55,56,57,58,59,60,61,62,63,64,65,66,67,68,69], fat [3,4,10,36,52,53,54,55,56,57,61,62,63,64,66,69,70,71,72,73,74,75,76,77,78,79,80,81], lipids [61,82], and main fatty acids [83] from the available literature are summarized in Figure 2.

Table 2 shows the values of the refractive index and absorption coefficient of biological tissues for frequencies from 0.5 to 2.0 THz.

### 3.2. Experimental Data

The temperature dependence of the spectrum absorption averaged over all samples and all measurements is shown in Figure 3. The scattering contribution can be estimated by applying Mie theory for spherical particles [84,85]. The approach essentially separates the independent contributions of true absorption and scattering losses, and thus determines the total extinction for different sizes of particles modelling various materials. However, in the THz range, scattering is not high in comparison to absorption for any tissue [85].

The refractive index temperature dependences for all samples and all measurements are shown in Figure 4. It should be noted that there are some discrepancies with the literature data (see Table 2), according to which the refractive index is about 1.6 in the THz region [54,86]. This difference can be attributed to the high water content of commercial pork fat. The effect of reflectance on an “air–tissue” boundary is excluded simply when we use relative measurements by dividing useful signals on a signal measured at reference conditions. The latter usually correspond to empty cuvette or tissue at initial conditions.

The THz spectra for the OD of H_2_O for layers of various thicknesses are shown in Figure 5, while the absorption spectrum of oleic acid at 28 °C is shown in Figure 6. The obtained spectra agree with the results presented by other authors [34,35,82]. The THz absorption spectra of adipose tissue and oleic acid, presented in Figure 2 and Figure 5, are practically similar in this spectral range, which is due to the fact that oleic acid predominates in porcine adipose tissue (see Table 2).

### 3.3. Molecular Modelling

Spatial configurations of the lowest energy conformers of five triglycerides of fatty acids (oleic, linoleic, palmitic, stearic, α-linolenic) are shown in Figure 7. Theoretical THz spectra of five triglycerides of fatty acids, taking into account their concentrations in porcine fat, are shown in Figure 8. The THz spectrum of porcine fat model, built using the supermolecular approach, is shown in Figure 9, and its interpretation is presented in Table 3. At the same time, only those vibrations that make a significant contribution to the formation of vibrational bands were taken into account.

To model the dehydration process as a generalized fat model, we used oleic acid triglyceride (Figure 7b). A confirmation of the correctness of the obtained local spatial configurations of intermolecular complexes (Figure 9) is the absence of negative values in the calculation of wave numbers. Almost all water molecules in our model are hydrogen bonded to the oleic acid triglyceride, with the exception of the water molecule, which acts as a binder between two water molecules forming hydrogen bonds with the carbonyl groups of the triglyceride (Figure 9c). This is due to the relatively large distance between the carbonyl groups, which does not allow the creation of a bound water dimer between them (Figure 9b). Taking this fact into account, the number of molecules in the first hydration shell was nine. For this case (Figure 9i), the number of formed hydrogen bonds was 15, and their length ranged from 1.8 to 2.8 angstroms.

A double-stranded linear complex consisting of 28 water molecules was chosen as a free water model (Figure 10). Due to the significant chain length, this model has a certain range of oscillations in the terahertz range. Therefore, the linear model is convenient to use when modeling the terahertz spectrum.

It should be noted that the percentages of bound water shown in Figure 9 and Figure 11 only apply to the selected fat model. If one selects a different model, these values will change. For example, for triglycerides of oleic and palmitic acids, these values decrease by approximately 2.5 times.

The model of fat dehydration process is shown in Figure 11. 

It can be seen from Figure 12 that the shape of optical density curves of fat (theoretical and experimental data) and oleic acid is similar, as evidenced by correlation analysis (Spearman’s rank correlation coefficient is equal to 1).

## 4. Discussion

Basing on the literature data for the absorption spectra of adipose tissue in the THz range, there are no pronounced bands in the range of 0.25–1 THz [34,57,88], which is consistent with the results of our study. Figure 3a shows the temperature dependence of absorbance at 1 THz. With temperature increasing, there is a decrease in the optical density, which is possibly explained by the decrease in scattering of the sample. Fat tissue typically consists of approximately 60–85% lipids and 15–30% water [43,89,90]. According to simple calculations, if absorption would decrease due to tissue dehydration, then absorption would decrease to 0.75 (and a decrease to 0.4 was obtained). 

In our studies, the temperature of the sample was slowly raised using a heating element, and the effect of terahertz radiation on the temperature change of the sample was not observed. The complex, inhomogeneous structure of adipose tissue, consisting of cells, septa, and capillaries with different thermal properties, can lead to inhomogeneous heat propagation during laser heating [91]. However, in our case, with slow heating, the propagation of heat in the adipose tissue could be considered homogeneous. 

Some inclusion into the OD decrease of the suppression of THz wave scattering at refractive index matching by free fatty acids released from adipocytes caused by thermally induced cell lipolysis (optical clearing effect) can be expected.

The fat cell size is in the range of 15–250 μm. The majority of the adipose tissue lipids are triglycerides. The size of its molecule, containing polyunsaturated fatty acids, is 1.5 nm. Triglyceride molecules can form various polymorphic forms. The most common forms are termed α, β’, and β in order of increasing melting point, packing density, and stability. The α form is the least stable and easily transforms to either the β’ form or the β form [88,92]. Adipose tissue can be represented as a quasi-ordered structure due to the crystal nature of triglycerides. Since quasi-ordered media have scattering properties from both random and ordered structures, it is important to account for even a small local order of particles when estimating the scattering properties. For the quasi-ordered structures, more comprehensive approaches, such as generalized Mie solution or T-matrix formalism, should be applied [85]. It was shown that crystal triglycerides are less than 40 μm in size in free form [88]. Sizes in the order of tens of micrometers are comparable in scale to the range of wavelengths in the THz range, and so the precise sizing of the crystals has a large effect on the optical properties of tissue in the THz range [93].

In addition, Figure 3 shows experimental THz spectra of the samples of porcine fat at different temperatures. It can be seen that with increasing temperature, the frequency of deformation vibrations of the chains of triglycerides of fatty acids increases. The explanation for this process may be as follows. Molecules of triglycerides of fatty acids are able to hold a certain number of water molecules on their surface using hydrogen bonds. These bonds are formed with oxygen atoms that are part of carbonyl groups and glycerol crosslinking of fatty acid triglycerides. When fatty acids are heated, the probability of breaking these hydrogen bonds increases. This leads to the fact that the weight of the chains decreases and their mobility increases. This, in turn, leads to an increase in the frequency of deformation vibrations of the chains of triglycerides of fatty acids.

In addition, the ability of fatty acids to retain water was considered when constructing a model of porcine fat. The influence of the amount of water associated with fatty acids on their theoretical THz spectra was investigated. We can hypothesize about the likely loss of water in a tissue sample at the beginning of heating (a decrease in the extinction coefficient due to a decrease in absorption). Then, free water first leaves the sample before bound water (see Figure 11). In this case, the non-monotonous behavior of the extinction coefficient on temperature is possible. And at the end of heating, the extinction coefficient decreases due to a decrease in scattering during immersion of the cells with the resulting fatty acids (lipolysis). The obtained simulation data (26.6% for bound water) are in good agreement with the literature data, according to which the maximum percentage of water in healthy adipose tissue is 30%, supposing that this is amount of bound and free water [43]. The model of fat with different ratios of free and bound water (see Figure 11) can be considered as a model for various pathological conditions of adipose tissue.

Because adipose tissue contains less water than muscle tissue, total body water tends to decrease with age. Older people have a higher percentage of body fat and are especially prone to dehydration. The proposed method for monitoring the water content in adipose tissue is objective in comparison with the traditional diagnosis of obesity and concomitant diseases. It is known that the ratio of lipids and water in tissues is a marker for diagnosing and monitoring inflammatory changes in adipose tissue at the cellular level in obesity, even when the standard body mass index is within the normal range [94].

The developed adipose tissue model can also be useful for predicting the level of obesity in diagnosing the risk of non-alcoholic fatty liver disease in people with obesity [95,96]. The proposed technique for measuring the water content in adipose tissue can be used on ex vivo biopsy material. Although biopsy is currently the gold standard for diagnosis, there is a need to increase the speed of analysis of tissue samples and use less invasive methods [97]. At the present time, rapid analysis of biopsy material with a reliable prediction can be implemented using terahertz spectroscopy and imaging [66]. In vivo, noninvasive studies of adipose tissue water content using a multimodal approach in combination with ultrasound, CT, and MRI will be the subject of our further research.

In general, despite the attractiveness, the methods of THz medical diagnosis are still far from practice [98]. The challenging problems of THz technologies, restraining their transfer to a clinical practice, are well-known and include the absence of robust contractions of waveguides for the THz-wave delivery to hardlyaccessible tissues and the limited depth of THz-wave penetration in biological tissues and liquids. Possible risks in the application of the proposed method caused by the stimulation of positive or negative biological effects in adipose tissues [99] can be avoided by choosing a proper THz intensity and exposure time. The proposed diagnostic approach is applicable in the case of taking a biopsy, when the excised tissue is examined, in the study of the surface layer of tissue in vivo by reflectance spectroscopy, and when using the immersion optical clearing method. Each of these methods reduces the effect of water absorption and contributes to an increase in the signal-to-noise ratio due to tissue dehydration, which, however, cannot always be well controlled.

## 5. Conclusions

The measurement and modeling of optical properties of adipose tissue and its components with temperature changes in the terahertz range were performed. The optical density of adipose tissue samples was shown to decrease as temperature increased, which can be associated mostly with the dehydration of the sample. Some inclusion into the optical density decrease of the suppression of THz wave scattering at refractive index matching by free fatty acids released from adipocytes caused by thermally induced cell lipolysis can be expected. Using complex molecular simulation of the adipose tissue at temperature change using classical molecular dynamics and quantum chemistry, we found correlations between the results of measurements and modeling. The exact percentage of different types of water (free and bound) in adipose tissue can be considered as a marker for diagnostics of such diseases as diabetes, obesity, and cancer.

## Figures and Tables

**Figure 1 diagnostics-12-02395-f001:**
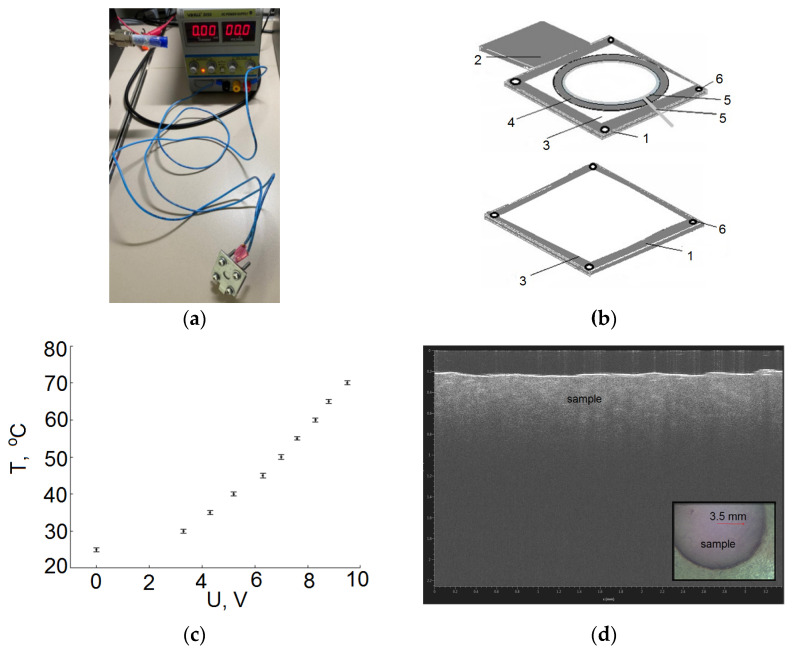
Prototype of a heating element used in experimental studies. (**a**) photo heating element connected with laboratory DC power supplies; (**b**) heating element diagram, where 1-metal base, 2-ledge plate, 3-fluoroplast, 4-washer, 5-wire, 6-screw; (**c**) calibration curve; (**d**) optical micrograph of the sample.

**Figure 2 diagnostics-12-02395-f002:**
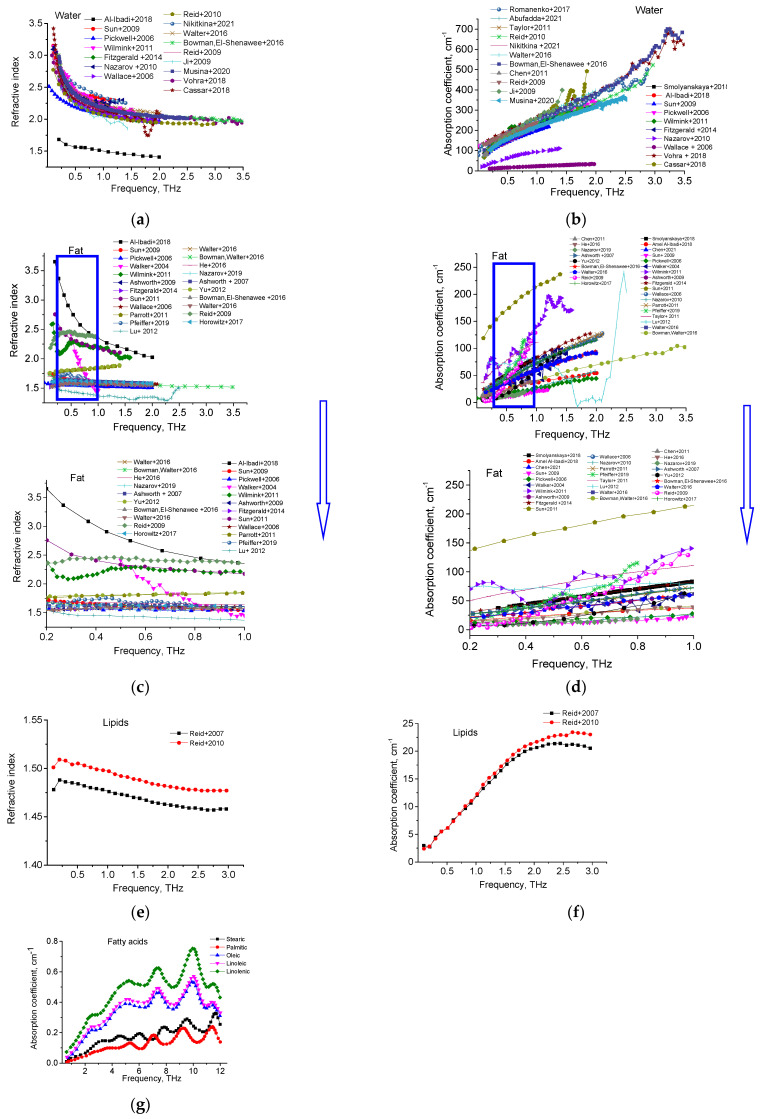
Refractive indices and absorption coefficients of water (**a**,**b**) [2,51,52,53,54,55,56,57,58,59,60,61,62,63,64,65,66,67,68,69], fat (**c**,**d**) [3,4,10,36,52,53,54,55,56,57,61,62,63,64,66,69,70,71,72,73,74,75,76,77,78,79,80,81], lipids (**e**,**f**) [61,82] and fatty acids (**g**) [83].

**Figure 3 diagnostics-12-02395-f003:**
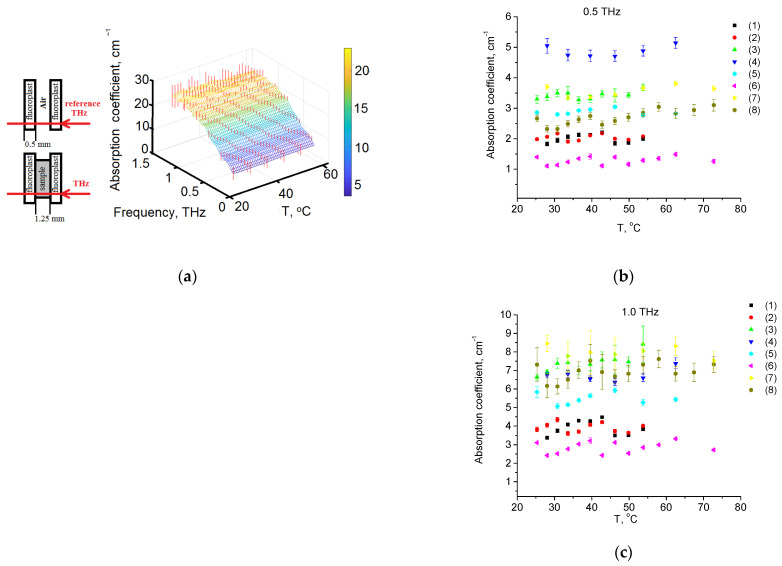
(**a**) Temperature dependence of the absorption spectrum in THz range; (**b**) for 0.5 THz; (**c**) for 1.0 THz. The measurements were carried out for eight samples (1)–(8).

**Figure 4 diagnostics-12-02395-f004:**
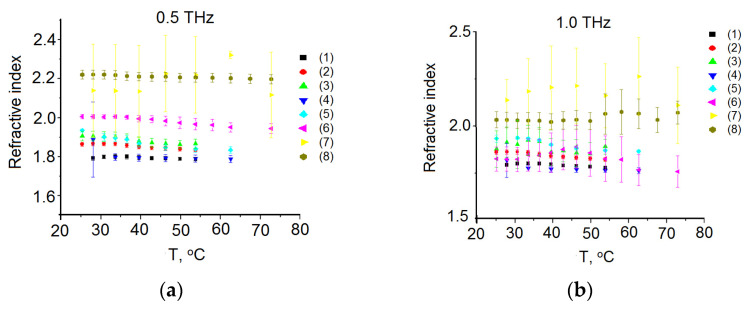
Temperature dependence of the refractive index (**a**) on 0.5 THz; and (**b**) on 1.0 THz. The measurements were carried out for eight samples (1)–(8).

**Figure 5 diagnostics-12-02395-f005:**
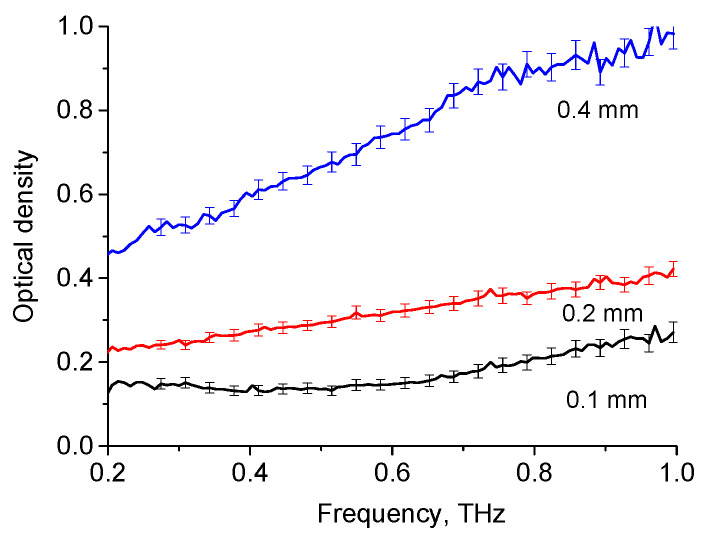
OD of liquid H_2_O for layers of various thicknesses in THz range.

**Figure 6 diagnostics-12-02395-f006:**
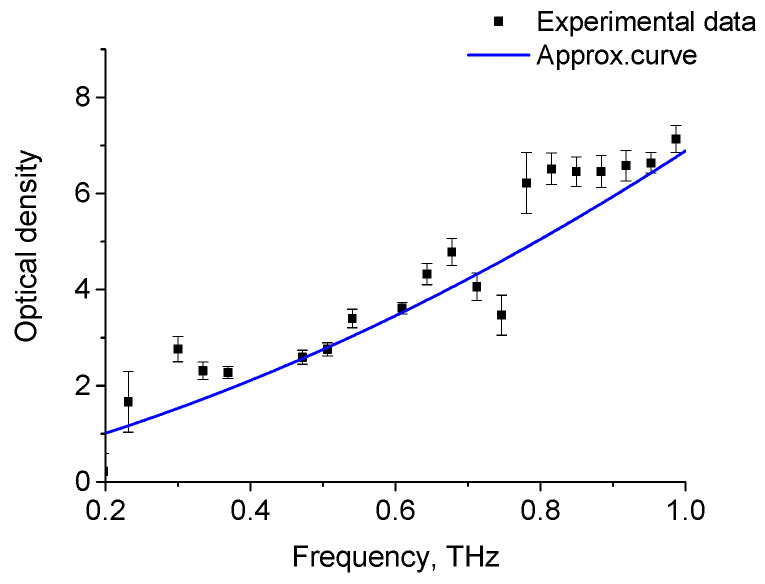
Absorption spectra of oleic acid at temperature of 28 °C. Experimental data-dots, approximation curve-line (y = 0.155 + 3.656x + 3.07x^2^).

**Figure 7 diagnostics-12-02395-f007:**
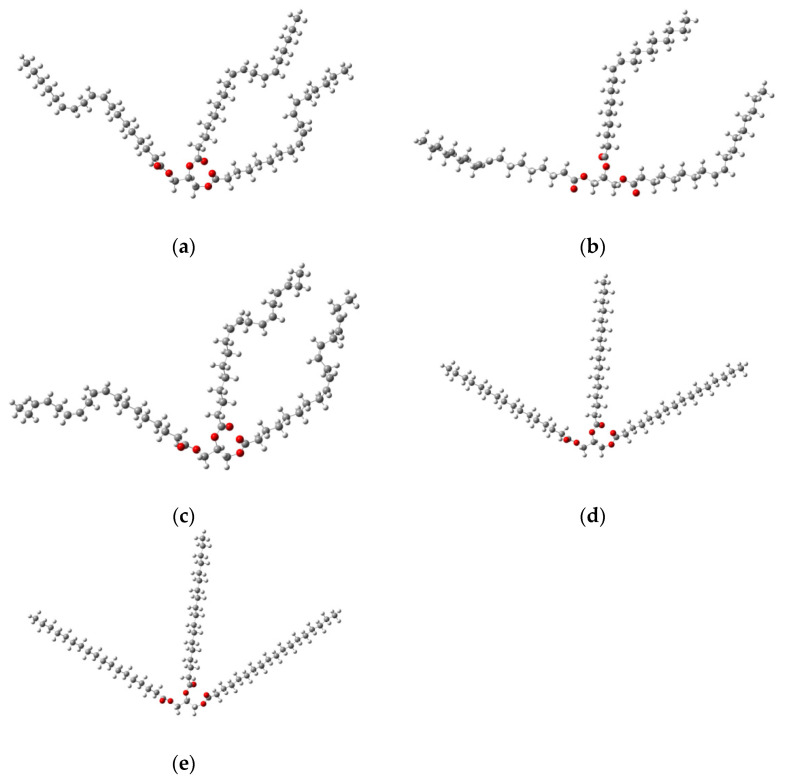
Spatial configurations of different triglycerides of fatty acids: linolic (**a**), oleic (**b**), α-linolenic(**c**), stearic (**d**) and palmitic (**e**) acids [87], where the white ball is a hydrogen atom; the red ball is an oxygen atom; and the gray ball is a carbon atom.

**Figure 8 diagnostics-12-02395-f008:**
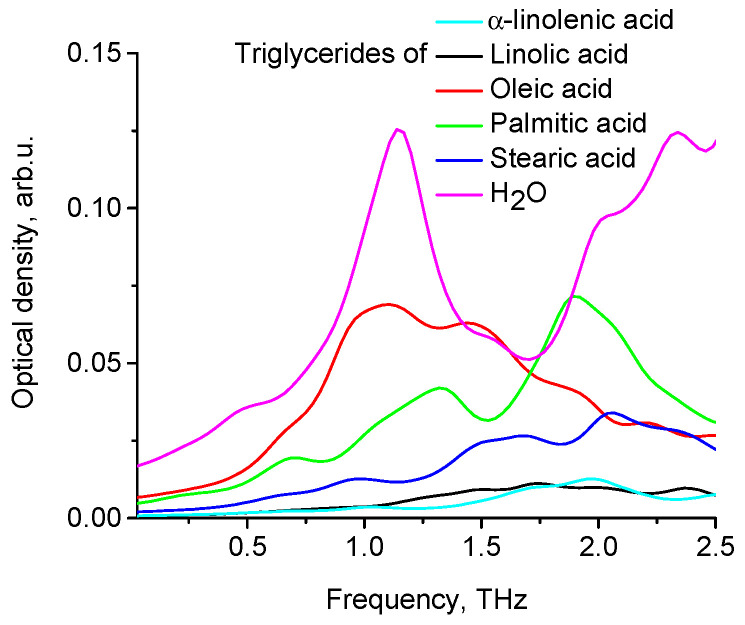
Absorption spectra of different triglycerides of fatty acids and H_2_O.

**Figure 9 diagnostics-12-02395-f009:**
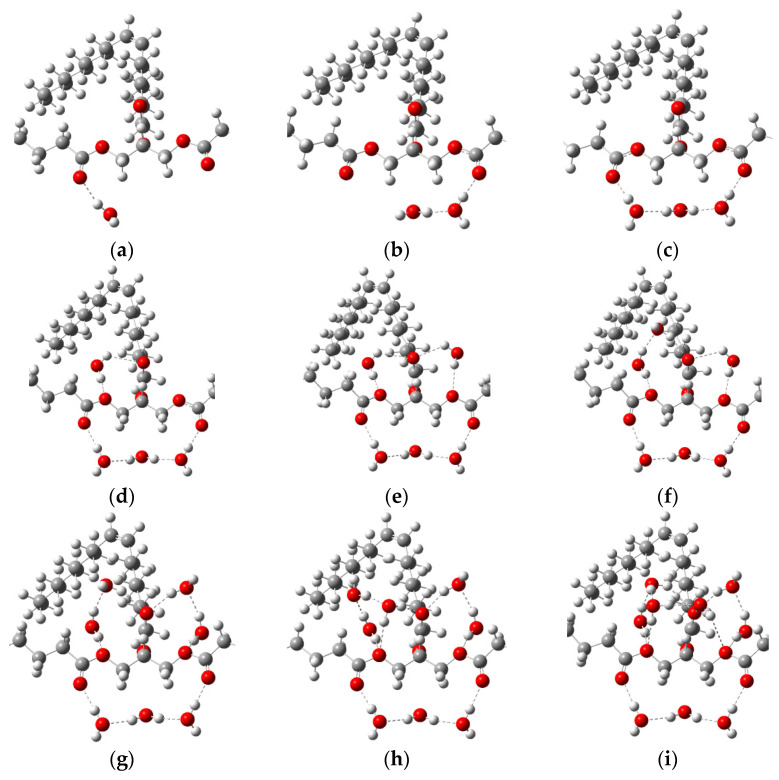
A model of fat with bound water in different percentages: (**a**) with one water molecule (2.0% by mass); (**b**) with two water molecules (3.9% by mass); (**c**) with three water molecules (5.8% by mass); (**d**) with four water molecules (7.5% by mass); (**e**) with five water molecules (9.2% by mass); (**f**) with six water molecules (10.9% by mass), (**g**) with seven water molecules (12.5% by mass), (**h**) with eight water molecules (14.0% by mass), (**i**) with nine water molecules (15.5% by mass). The dotted lines show hydrogen bonds. The white ball is a hydrogen atom; the red ball is an oxygen atom; and the gray ball is a carbon atom.

**Figure 10 diagnostics-12-02395-f010:**
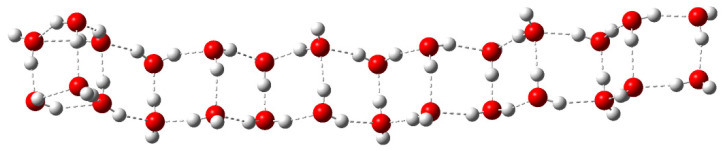
Linear structure of 28 water molecules, where the white ball is a hydrogen atom; the red ball is an oxygen atom. The dotted lines show hydrogen bonds.

**Figure 11 diagnostics-12-02395-f011:**
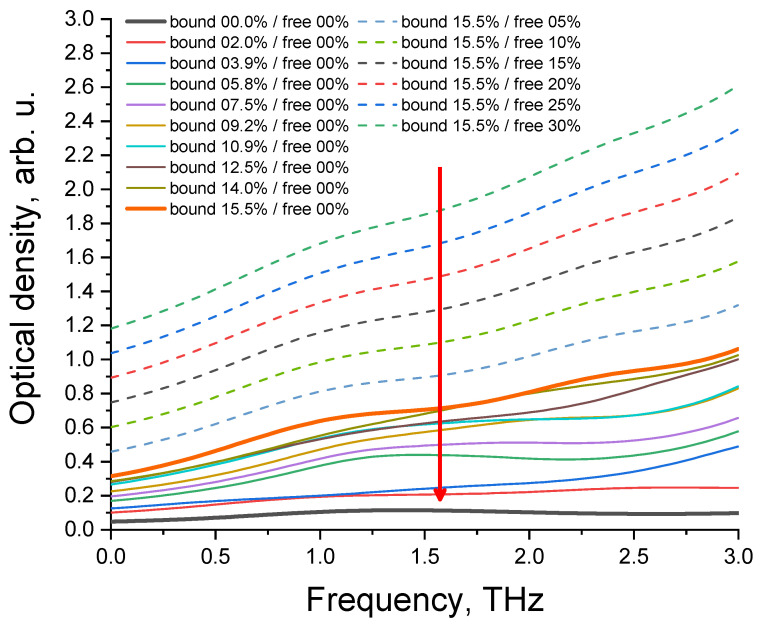
Absorption spectra of model fat with different ratios of free and bound water. The red arrow indicates the direction of the change in the percentage ratio between free and bound water in fat during dehydration.

**Figure 12 diagnostics-12-02395-f012:**
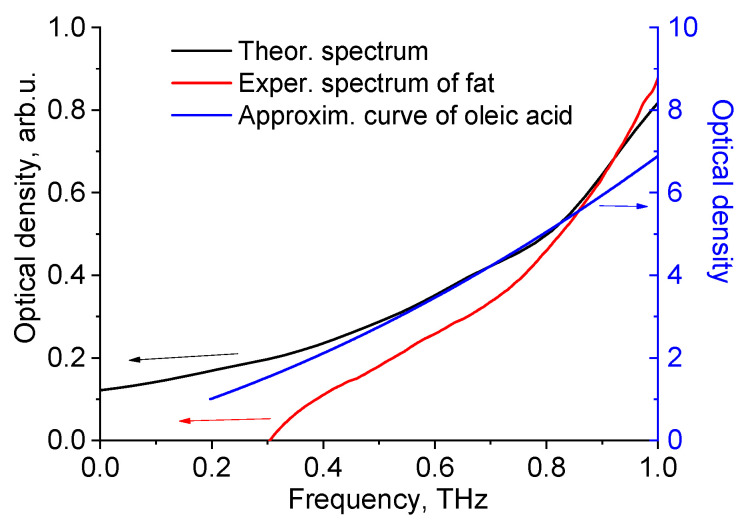
Theoretical and experimental THz spectra of porcine fat and approximation curve for experimental data of oleic acid (from Figure 8).

**Table 1 diagnostics-12-02395-t001:** Melting temperatures of porcine fat (triglycerides) and free fatty acids (FFA) and their concentrations in % by mass [48].

Porcine Fat (in Solid State)	Melting Temperature, °C	FFA, Melting Temperature, °C (Concentration, % by Mass)				
		*Palmitic*	*Stearic*	*Oleic*	*Linoleic*	*Linolenic*
Triglycerides	36–45	63 (27%)	70 (14%)	16 (45%)	−5 (5%)	−11 (5%)

**Table 2 diagnostics-12-02395-t002:** The refractive index and absorption coefficient of main component of adipose tissue for frequencies 0.5, 1, 1.5 and 2 THz.

Object	Refractive Index				References	Comments
	Absorption Coefficient, cm^−1^					
	*0.5 THz*	*1 THz*	*1.5 THz*	*2 THz*		
Water	2.4 157.8	2.2 220.3	2.1 270.6	2.0 316.9	[51] *	Water samples were laboratory grade purified and deionized, and studied at room temperature and atmospheric pressure
	2.3 16.5	2.1 23.4	2.0 28.9	2.0 33.6	[23]	
	2.2 187.9	2.1 253.6	2.0 299.1	2.0 339.0	[54]	
	2.2 181.8	2.0 236.3	2.0 284.9	1.9 343.6	[60]	
	2.4 200.6	2.2 269.4	2.1 329.3	2.1 396.8	[61]	
	2.3 189.4	2.1 272.3	2.0 328.5	2.0 395.1	[58] *	Distilled at room temperature and normal pressure
	2.3 196.6	2.16 272.8	2.1 326.1	2.0 396.6	[62]	
	2.3 163.2	2.1 255.5	2.0 348.5	2.1 -	[59] **	Liquid state
Fat	1.6 36.3	1.6 63.3	1.5 80.0	1.5 93.5	[3]	Human
	1.6 44.5	1.6 72.5	1.6 98.7	1.6 123.3	[23]	
	2.7 27.1	2.3 37.1	2.2 46.3	2.0 54.3	[52]	
	1.6 37.2	1.6 59.7	1.6 81.7	1.5 90.7	[80]	
	1.6 42.6	1.6 75.5	1.6 100.7	1.6 115.6	[81]	
	1.5 14.8	1.5 28.5	1.5 37.8	1.5 44.4	[54]	Human, ex vivo
	1.6 46.5	1.6 73.6	1.6 98.8	1.6 115.4	[77]	Human, at room temperature
	1.6 38.7	1.6 63.3	1.6 78.0	1.5 92.7	[61]	Fresh
	1.4 68.7	1.4 78.6	1.3 41.2	1.3 6.1	[75]	Pork
	1.6 36.3	1.6 60.7	1.6 80.5	1.6 93.7	[79]	Fresh, bovine
Lipids	1.5 6.1	1.5 12.3	1.5 18.3	1.5 21.7	[60]	Beef tallow, pure lipids from beef
	1.5 6.1	1.5 12.0	1.5 17.6	1.5 20.7	[82]	Lard, pure commercially available lipid

* Additional data provided by the authors. ** More detailed data are not available.

**Table 3 diagnostics-12-02395-t003:** Interpretation of the theoretical THz spectra of porcine fat in the frequency range from 0 to 2.5 THz.

Frequency (THz)	Interpretation
0.72	Deformation (bending) oscillation of the left chain of palmitic acid triglyceride
1.20	Deformation (torsion) oscillation of the left chain of oleic acid triglyceride
1.41	Deformation (torsion) oscillation of the central chain of palmitic acid triglyceride and mixed deformation (torsion) oscillation of the side chains of oleic acid triglyceride
1.92	Deformation (bending) oscillation of the right chain of palmitic acid triglyceride and a similar oscillation of the central chain

## Data Availability

The data presented in this study are available on request from the corresponding author. The data are not publicly available due to privacy or ethical restrictions.

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
