# Peer review of "Measurement and Modeling of the Optical Properties of Adipose Tissue in the Terahertz Range: Aspects of Disease Diagnosis"

_diagnostics, 2022, doi:10.3390/diagnostics12102395_

Round 1

Reviewer 1 Report

The manuscript diagnostics-1863487 has been devoted to present numerical and experimental results related to optical properties in the terahertz range of adipose tissue and its components with influence of temperature. Please see below a list of comments to the authors: 1. The authors wrote “The thickness of the samples were about 1.5 mm, their area 144 was amounted to 1 cm2.” How was prepared this cut in order to guarantee an error in thickness of 10 μm? 2. I like the photo shown in figure 1a; but if possible please provide an optical micrograph of the sample. 3. In table 2, a column 6 describes the water as: liquid, pure or distilled with no clear distinction. An edition is required in order that this information can be useful. 4. It is not clear the role of the scattering and reflection in the data plotted in Figures 3-5. A description with better details and correction of required is mandatory. 5. Propagation of temperature in biological samples is fractional and inhomogeneous; then, the optical absorption should be also anisotropic and dependent on incident polarization. The authors are invited to discuss about this issue considering for instance: https://doi.org/10.1016/j.ijthermalsci.2022.107734 6. It is suggested to confront the advantages and disadvantages of the proposed technique with other methods for disease diagnosis. You can see for instance: http://dx.doi.org/10.1088/2040-8986/ab4dc3 7. Keywords should be separated with semicolon and a proofreading is mandatory. 8. Error bar in experimental data should be included. 9. Collective citations are suggested to be split in order to better justify each reference selected to present the panoramic of this topic. 10. A color bar in figure 3a is missing?

Author Response

Response to Reviewer 1 Comments

Point 1: The authors wrote “The thickness of the samples were about 1.5 mm, their area 144 was amounted to 1 cm2.” How was prepared this cut in order to guarantee an error in thickness of 10 μm?

Response 1: Thank you for comment. The text describing the method of preparation of sections has been changed: “In advance, samples of adipose tissue frozen at a temperature of -20℃ were cut into pieces with a thickness of slightly above 2 mm. Then, using a cylindrical punch with an inner diameter of 9 mm, a cylindrically shaped sample was cut. This sample was thawed, placed inside a metallic holderwith a fixed height of 1.25 mm, and excess adipose tissue was removed by a scalpel.”; “Metallic holder height measurements were measured using micrometer "MK 0-25 mm", model 102 (Plant "Caliber", Russia).”

Point 2: I like the photo shown in figure 1a; but if possible please provide an optical micrograph of the sample.

Response 2: We have added a photo of the optical micrograph of the sample (see Fig.1c).

Point 3: In table 2, a column 6 describes the water as: liquid, pure or distilled with no clear distinction. An edition is required in order that this information can be useful.

Response 3: Thank you, we have added the required information to table.

  1. It is not clear the role of the scattering and reflection in the data plotted in Figures 3-5. A description with better details and correction of required is mandatory.

Response 4: We have added the requested information to the manuscript.

“The scattering contribution can be estimated by applying Mie theory for spherical particles [85, 86]. The approach essentially separates the independent contributions of true absorption and scattering losses and thus determines the total extinction for different sizes of particles modelling various materials. However in THz range, scattering is not high in comparison with absorption for any tissue [86].”; “The effect of reflectance on an "air-tissue" boundary is excluded simply when we use relative measurements by dividing useful signals on a signal measured at reference conditions. The latter usually correspond to empty cuvette or tissue at initial conditions.”

  1. Propagation of temperature in biological samples is fractional and inhomogeneous; then, the optical absorption should be also anisotropic and dependent on incident polarization. The authors are invited to discuss about this issue considering for instance: https://doi.org/10.1016/j.ijthermalsci.2022.107734.

Response 5: Thank you, we have taken into account the reviewer's comment and made chages in the MS. Discussion of the polarization properties of adipose tissue is beyond the scope of this study. However, one can immediately say that due to the specific cellular structure of adipose tissue, the polarization anisotropy should not be significant, especially in the terahertz range (see for example V. V. Tuchin, “Polarized light interaction with tissues,” J. Biomed. Opt. 21(7), 071114-1-37 (2016)).

“In our studies, the temperature of the sample was slowly raised using a heating element, the effect of terahertz radiation on the temperature change of the sample was not observed. The complex inhomogeneous structure of adipose tissue, consisting of cells, septa and capillaries with different thermal properties, can lead to inhomogeneous heat propagation during laser heating [92]. However, in our case, with slow heating, the propagation of heat in the adipose tissue could be considered homogeneous”

  1. It is suggested to confront the advantages and disadvantages of the proposed technique with other methods for disease diagnosis. You can see for instance: http://dx.doi.org/10.1088/2040-8986/ab4dc3

Response 6: We have added the following information to the manuscript.

“In general, despite the attractiveness,the methods of THz medical diagnosis are still far from medical practice [99]. The challenging problems of THz technologies, restraining their transfer to a clinical practiceare well-known and include the absence of robust constractions of waveguides for the THz-wave delivery to hardly-accessible tissues and the limited depth of THz-wave penetration in biological tissues and liquids. The possible risks in the application of the proposed method caused by stimulation positive or negative biological effects in adipose tissues [100] can be avoided by choosing a proper THz intensity and exposure time. The proposed diagnostic approach is applicable in the case of taking a biopsy, when the excised tissue is examined, in the study of the surface layer of tissue in vivo by reflectance spectroscopy, and also when using the immersion optical clearing method. Each of these methods reduces the effect of water absorption and contributes to an increase in the signal-to-noise ratio due to tissue dehydration, which, however, cannot always be well controlled.”

  1. Keywords should be separated with semicolon and a proofreading is mandatory.

Response 7: Thank you, fixed.

  1. Error bar in experimental data should be included.

Response 8: Thank you, fixed.

  1. Collective citations are suggested to be split in order to better justify each reference selected to present the panoramic of this topic.

Response 9: We have taken into account the reviewer's comment.

  1. A color bar in figure 3a is missing?

Response 10: Thank you, fixed.

Reviewer 2 Report

In this paper, the authors have implemented THz spectroscopy and DFT to study the optical properties of adipose tissue. Importantly, the authors employ the structural model of butanoic acid triglyceride as the adipose tissue theory model to study the dehydration process caused by temperature increase. This research is significant for diagnosing and monitoring diseases by confirming the exact percentage of free and bound water in adipose tissue. Additionally, the description and explanation of this manuscript are not clear at several points. So, this manuscript should be major revision before it can be accepted. Following are the comments should be considered by the authors.

1.     In the introduction part, the applications of THI have less relationship with the purpose of this manuscript, the authors should highlight the importance on THS.

2.     The specific parameters of T-SPEC terahertz spectrometer are not indicated. In addition, whether nitrogen protection is employed during the THz measurement.

3.     The structural model of oleic acid is better than butanoic acid triglyceride as the porcine fat model, because its percentage in adipose tissue is the highest (45%). Thus, the authors should use the structural model of oleic acid to simulate the dehydration process.

4.     In Figure 13, the experimental THz spectrum of porcine fat need to add error bar if eight samples were measured. In addition, the absorption coefficient of oleic acid should be converted to optical intensity, it is easy to compare the experimental and theory spectra of the samples.

5.     The spaces between words need to be added at Line 64-65. The authors should check the format of the manuscript throughout.

6.     The legend is not marked in Figure 3 (a).

7.     In Figure 6, the fitting formula and correlation coefficient are not exhibited.

Author Response

Response to Reviewer 2 Comments

Point 1: In the introduction part, the applications of THI have less relationship with the purpose of this manuscript, the authors should highlight the importance on THS.

Response 1: We have added the following information to the manuscript.

“THz signals with wavelength of approximately 300 µm provide a good balance between penetration, deepness and spatial resolution to identify a molecular biomarker spatial distribution in tissue [13].

Biomarker detection can be helpful in the diagnosis of cancer. As different substances have different THz spectra (i.e., spectral “fingerprints”), THz spectroscopy (THS) can be applied to realize fast and accurate identification of biomarkers in cancer tissue. Furthermore, THS can be combined with various algorithms to realize the quantitative analysis of cancer biomarkers, which may be a potential tool for rapid cancer staging [14]. However, in most cases, various substances (such as water, proteins, lipids, and other organic components) are also present in the tissue, and cancer biomarker concentration is usually very low. These conditions result in a low spectral signal-to-noise ratio (SNR), which gives rise to absorption peaks that cannot be readily identified [15]. Therefore, current THz-based studies of biomarkers are mostly done on pure biomarkers. If we can construct THz images based on the resonance peaks of biomarkers, cancerous areas can be identified more accurately. However, further development is required to realize this objective as current imaging systems with high-power continuous-wave THz sources can only generate THz waves at a fixed frequency, and pulsed THz sources cannot provide enough spectral power. Thus, both THI and THS are limited by source energy, leading to poor SNR in the spectral results. To address this issue, some researchers are working to enhance the spectral SNR, which is expected to help detection of biomarkers in mixed samples.”

Point 2: The specific parameters of T-SPEC terahertz spectrometer are not indicated. In addition, whether nitrogen protection is employed during the THz measurement.

Response 2: We have added an specific parameters of T-SPEC terahertz spectrometer: “A photoconductive antenna illuminated by the ultrashort laser pulses, was used for generation ofTHz radiation and itsdetection. The pumping laser provided pulses of 10-150 fs at 1050±40 nm wavelength with power of about100 mW and 30-100 MHz pulse repetition rate. For more efficient collimation and focusing of THz radiation, a substrate lens fabricated from high resistance silicon was attached to the backside of each antenna.Thesample holder was placed in the optical path. Atmospheric air was in the optical path, and the cover of the device was open to ensure the outflow of excess heat from the outer part of the sample holder. There were no advantages in using the nitrogen in the range from 0.2 to 1 THz compared to atmospheric air.”

Point 3: The structural model of oleic acid is better than butanoic acid triglyceride as the porcine fat model, because its percentage in adipose tissue is the highest (45%). Thus, the authors should use the structural model of oleic acid to simulate the dehydration process.

Response 3:In connection with the possibility of using a faster computer, the generalized molecular model of fat, when considering the dehydration process, was changed to a more adequate one. Instead of butanoic acid triglyceride, oleic acid triglyceride was used.

“…To model the dehydration process as a generalized fat model, we used oleic acid triglyceride (Fig. 7(b)). A  confirmation of the correctness of the obtained local spatial configurations of intermolecular complexes (Fig. 9) is the absence of negative values in the calculation of wave numbers. Almost all water molecules in our model are hydrogen bonded to the oleic acid triglyceride, with the exception of the water molecule, which acts as a binder between two water molecules forming hydrogen bonds with the carbonyl groups of the triglyceride (Fig. 9(c)). This is due to the relatively large distance between the carbonyl groups, which does not allow the creation of a bound water dimer between them (Fig 9(b)). Taking this fact into account, the number of molecules in the first hydration shell was nine. For this case (Fig. 9(i)) the number of formed hydrogen bonds was 15, and their length ranged from 1.8 to 2.8 angstroms.”; “A double-stranded linear complex consisting of 28 water molecules was chosen as a free water model (Fig. 10). Due to the significant chain length, this model has a certain range of oscillations in the terahertz range. Therefore, the linear model is convenient to use when modeling the terahertz spectrum.”

Point 4: In Figure 13, the experimental THz spectrum of porcine fat need to add error bar if eight samples were measured. In addition, the absorption coefficient of oleic acid should be converted to optical intensity, it is easy to compare the experimental and theory spectra of the samples.

Response 4: Thank you, fixed

Point 5:The spaces between words need to be added at Line 64-65. The authors should check the format of the manuscript throughout.

Response 5: Thank you, fixed

Point 6: The legend is not marked in Figure 3 (a).

Response 6: Thank you, fixed

Point 7: In Figure 6, the fitting formula and correlation coefficient are not exhibited.

Response 7: Thank you, fixed

Reviewer 3 Report

This manuscript is devoted to measuring and modeling of optical properties in the terahertz frequency range of adipose tissue. The review of investigation of fatty tissues, water and lipids content in tissues are presented in manuscript. The investigation of process of fatty tissues dehydratation at heating in THz frequency range was carried out. The model of porcine fat as combination of five triglycerides of fatty acids was considered. The results of measuring the optical characteristics of porcine fat and comparison with model data for individual fatty acids, water and literature data are demonstrated. These results can be interesting for scientific groups in areas of development of THz sensors and of medical diagnostics. 

There are some points to correct or to make the information more clear:

1) It is necessary to explain abbreviation at first appearance in text or figures. E.g.: MR (54th line), CT and MRI (365th line). But sometimes the abbreviation is explained twice (optical density (OD) is explained in abstract (20th line) and in 151st line.

2) The numbers in the figure 1 (b) are very small. Besides it will be more convenient to add the explaining these numbers presented in text to the Figure caption. Besides that the number of 7 in text is presented twice (for the wire spiral with tape and for screws).

3)  The comments in Table 2 for water are “liquid”, “pure” “distilled”. It is not clear what the aggregative states of pure and distilled water are in that references. It is more convenient to combine the lines with the same comments in series and similarly for fat.

4) There is a large spread in values for fat (Figure 2 (c) and (d)). Are the dependences presented here for various types of fatty tissues? It is possible to be more convenient to present here (as inset, possibly) the low frequency part for pork fat because there is the comparison the fig 5 with fig 2 (d) further.

5) There are many places where the spaces are skipped (“provideusefulinformationfordifferentiationofhealthyandcanceroustissues” in 64-65th lines; “linoleicacids” in 109th line; “[43].For” in 110th line; “70-80%.The” in 121st line; “Table2.Therefractiveindex” in 215th line etc.)

Author Response

Response to Reviewer 3 Comments

Point 1: It is necessary to explain abbreviation at first appearance in text or figures. E.g.: MR (54th line), CT and MRI (365th line). But sometimes the abbreviation is explained twice (optical density (OD) is explained in abstract (20th line) and in 151st line

Response 1: We have taken into account the reviewer's comment.

Point 2: The numbers in the figure 1 (b) are very small. Besides it will be more convenient to add the explaining these numbers presented in text to the Figure caption. Besides that the number of 7 in text is presented twice (for the wire spiral with tape and for screws).

Response 2: We have added explaining the numbers presented in text to the Figure 1b caption. We have made the designations of the numbers in the figure large. We removed the double notation

Point 3:The comments in Table 2 for water are “liquid”, “pure” “distilled”. It is not clear what the aggregative states of pure and distilled water are in that references. It is more convenient to combine the lines with the same comments in series and similarly for fat

Response 3: We have added the requested information to the manuscript.

  1. There is a large spread in values for fat (Figure 2 (c) and (d)). Are the dependences presented here for various types of fatty tissues? It is possible to be more convenient to present here (as inset, possibly) the low frequency part for pork fat because there is the comparison the fig 5 with fig 2 (d) further.

Response 4: We have added low-frequency regions in Figure 2 (c) and 2 (d) as insets.

  1. There are many places where the spaces are skipped (“provideusefulinformationfordifferentiationofhealthyandcanceroustissues” in 64-65th lines; “linoleicacids” in 109th line; “[43].For” in 110th line; “70-80%.The” in 121st line; “Table2.Therefractiveindex” in 215th line etc.)

Response 5: Thank you, fixed.

Round 2

Reviewer 1 Report

In my opinion, the authors have importantly improved the presentation of their work, and now, their manuscript can be considered for publication as it is.

Reviewer 2 Report

The authors have made an effort to revise their manuscript according the comments.